# Current Perspective of Sialylated Milk Oligosaccharides in Mammalian Milk: Implications for Brain and Gut Health of Newborns

**DOI:** 10.3390/foods10020473

**Published:** 2021-02-21

**Authors:** Madalyn Hobbs, Marefa Jahan, Seyed A. Ghorashi, Bing Wang

**Affiliations:** 1Graham Centre for Agricultural Innovation, Charles Sturt University, Wagga Wagga, NSW 2678, Australia; madalyn.h@hotmail.com (M.H.); mjahan@csu.edu.au (M.J.); aghorashi@csu.edu.au (S.A.G.); 2School of Animal & Veterinary Sciences, Charles Sturt University, Wagga Wagga, NSW 2678, Australia

**Keywords:** human milk oligosaccharides, sialic acid, health, cognition, newborn

## Abstract

Human milk oligosaccharides (HMOs) are the third most abundant solid component after lactose and lipids of breast milk. All mammal milk contains soluble oligosaccharides, including neutral milk oligosaccharides (NMOs) without sialic acid (Sia) moieties and acidic oligosaccharides or sialylated milk oligosaccharides (SMOs) with Sia residues at the end of sugar chains. The structural, biological diversity, and concentration of milk oligosaccharides in mammalian milk are significantly different among species. HMOs have multiple health benefits for newborns, including development of immune system, modification of the intestinal microbiota, anti-adhesive effect against pathogens, and brain development. Most infant formulas lack oligosaccharides which resemble HMOs. Formula-fed infants perform poorly across physical and psychological wellbeing measures and suffer health disadvantages compared to breast-fed infants due to the differences in the nutritional composition of breast milk and infant formula. Of these milk oligosaccharides, SMOs are coming to the forefront of research due to the beneficial nature of Sia. This review aims to critically discuss the current state of knowledge of the biology and role of SMOs in human milk, infant formula milks, and milk from several other species on gut and brain health of human and animal offspring.

## 1. Introduction

Up until the point of weaning, the milk is the most important, and often sole, source of nutrients for the neonate. Breast-milk is considered to be the gold-standard in infant nutrition and often called ‘living tissue’, as it contains numerous components associated with a wide range of health benefits, such as reduced incidence and severity of diarrheal infections [1,2], otitis media, respiratory illnesses, gastrointestinal illnesses, and better neurocognitive performance [3,4,5,6,7,8]. Breastfed children are less likely to be overweight or obese and less prone to type 2 diabetes later in life [9,10]. Breastmilk is a critical source of energy and nutrients during illness and reduces mortality among malnourished children [11]. Breastmilk has also been linked to higher intelligence in later childhood [11]. Many nutrients in human milk may support these health benefits. One promising nutrient is human milk oligosaccharides (HMOs), which is the third most abundant component of human breast milk after lactose and lipids [12,13,14]. Human milk oligosaccharides are reported to benefit the neonate in infection protection, brain maturation, growth, and development of infants, and microbiome establishment and maintenance [15]. Many studies have indicated that breast-fed babies have a better gut health and microbiota content compared to formula-fed babies due to the higher diversity and concentration of oligosaccharide in breast milk [1,16,17].

The complex free oligosaccharides are made up of a core molecule consisting of lactose (involving glucose (Glu) linked to galactose (Gal) by a β1, 4-glycosidic bond) to which other monosaccharides, such as fucose (Fuc), N-acetylglucosamine (GlcNAc), N-acetylgalactosamine (GalNAc), and sialic acid (Sia), can be attached [12,18]. All HMOs carry lactose (Galβ1-4Glc) at the reducing end, which can be elongated in β1-3- or β1-6-linkage by two different disaccharides, either Galβ1-3GlcNAc (type 1 chain) or Galβ1-4GlcNAc (type 2 chain). Over 200 different oligosaccharides have been characterized in human milk [19,20], varying from 3 to 22 simple sugars [21]. According to Urashima, et al. [22], 247 diversities of HMOs have been separated, of which 162 chemical structures have been characterized. Human breast milk has a far greater diversity and quantity of oligosaccharides than other reported species, with about 50–70% being neutral HMOs, commonly fucosylated (containing fucose) and non-fucosylated oligosaccharides; and 10–30% being acidic HMOs containing one or more Sia residues, so-called sialylated milk oligosaccharides (SMOs) [14,23]. Sialylated milk oligosaccharides are one of essential conditional bioactive component for brain and cognitive development of newborns. In addition, SMOs potentially act as a prebiotic for probiotic bacteria in the gut environment [24], making studies of these SMOs critical for future developments in human and animal health.

Human milk contains significant amount of oligosaccharides at concentration of 20–23 g/L in colostrum and 12–14 g/L in mature milk [25]. However, the soluble oligosaccharides are not unique to human milk, with oligosaccharides being found in the milk of nearly all mammals [26]. The quantity and diversity of milk oligosaccharides varies between species. Most infant formulae are bovine-based, but the abundance and composition of bovine milk oligosaccharides are different from HMOs [27]. Although galacto-oligosaccharides (oligomers of galactose with 3 to 10 galactose unit and glucose at the reducing end) (GOS) and fructo-oligosaccharides (oligomers of fructose with 2 to more than 60 fructose unit and glucose at the reducing end) (FOS) [20] are currently included in some infant milk products to mimic the biological effects of HMO based on their structural similarities to the core molecules of HMOs, both GOS and FOS are not present in human milk [28]. In addition, the concentration and diversity of HMOs are significantly lacking in the current standard infant formula. In particular, SMOs are left out of the infant formula [12,29,30,31,32] because of the unavailability of food grade SMOs in the market for manufacturing infant formula.

### 1.1. Sialic Acid

Sia, a family of over 50 naturally occurring nine-carbon acidic monosaccharides, is a biomarker of SMOs, sialylated glycoprotein, and sialylated glycolipids. Currently, more than 15 structurally distinct Sias have been identified in humans [33,34,35]. These acidic monosaccharides are found at the outermost end of glycan chains and decorate all cell surfaces and soluble proteins in higher animals and some microorganisms [36]. In human milk, however, Sias are predominantly conjugated to oligosaccharides (approximately 69–80%) followed by glycoprotein (~ 20–30%) and glycolipids (approximately 0.2–1%) [20,37,38]. The free form Sia is approximately 2–3% of the total milk Sia [37]. The Sia family can be classified and divided into N-acetylneuraminic acid (Neu5Ac), the most predominant form of Sia in human cells and many mammalian cells, and the non-human Sia N-glycolylneuraminic (Neu5Gc) [18]. Both Neu5Ac and Neu5Gc are the most common Sia forms in nature [39]. Another new member of the Sia family is 2-keto-3-deoxy-D-glycero-D-galacto-nononic acid (KDN), which is often expressed at lower levels than Neu5Ac and Neu5Gc in mammalian tissues. Sia is rarely found in its free form in nature, instead attaching at the terminal, non-reducing end of chains of complex oligosaccharides in glycolipids, glycoproteins, and mucin. Sia as sialylated glycoconjugates can be expressed in nearly all vertebrate cell types, tissues, and body fluids, including milk, plasma, saliva, urine, and cerebrospinal fluid [40]. The highest concentration of Sia is expressed in the gangliosides and glycoproteins of the central nervous system [35,41] that play an essential role in the transmission and storage of information in the brain [35].

Sia is found in variable concentrations in human breast milk with concentrations of approximately 5.04 mmol/L in colostrum and 1.98 mmol/L in mature milk of full-term mother [35]. However, in the case of a pre-term mother, the concentration of Sia is 5.76 mmol/L and 2.56 mmol/L in colostrum and mature milk, respectively [35]. Thus, pre-term human milk contains ~10–20% higher levels of Sia than full-term mother milk. The concentration of Sia in human milk varies between different ethnic groups and different methods for milk collection and methods used for Sia analysis [37,38]. The concentration of Sia decreases during the course of lactation, with Sia being found at three-fold greater levels in colostrum than in mature milk. Furthermore, Sia concentration in transitional milk is lower than colostrum and higher than mature milk [38]. This downward trend was also observed in both oligosaccharide-conjugated and glycoprotein-conjugated Sia and free Sia [38]. Overall, approximately 57% of N-glycans and most glycolipids in human milk are sialylated glycoconjugates [20], which are involved in numerous developmental processes; for instance, they work as building block of brain gangliosides and sialoglycoproteins and have immunomodulatory and immunostimulatory properties [42] and neurodevelopment and cognition [18,35].

### 1.2. Sialylated Milk Oligosaccharides in Human Breast Milk

Human milk is a rich source of soluble SMOs. More than 55 structurally distinct SMOs have been characterized so far [43]. The simplest and predominant SMOs present in human milk are trisaccharides, 6′sialylactose (SL), followed by 3′SL [44], which are formed by the binding of a Sia by α2,6- and/or α2,3-glycosidic linkages to the terminal galactose unit of lactose in the 3 and 6 position, respectively [37,45]. The mean concentration of 3′-SL and 6′-SL in term human milk during the first 100 days of lactation was 0.19 and 0.64 g/L, respectively [46]. The other SMOs in human breast milk include sialyllacto-N-tetraoses (LST), such as LSTa, LSTb, and LSTc, at a concentration level of 0.06, 0.13, and 0.25 g/L, respectively; and disialyllacto-N-tetraose (DSLNT), at levels of 0.50 g/L, during the 2nd to 120 days of lactation, respectively [43]. Following the same trends of total Sia, the concentration of 6′SL in human milk also declines during the course of lactation, with the highest concentration found in the first month at level of 0.50–0.64 g/L compared to 0.25 g/L in the second month of lactation [44]. Interestingly, the concentration of 3′SL remains relatively consistent throughout lactation at approximately 0.22 g/L [44]. There are limited data available on comparison of SMOs concentration in different ethnic group and geographic human milk. However, the concentration of 3′SL and 6′SL in human milk varies between different study groups, ranging from 0.2 (0.1–0.3) g/L for 3′SL and 0.5 (0.2–1.22) g/L for 6′SL in mature milk [47]. In pre-term human milk, the concentration of 3′-SL and 6′-SL were about 0.24 and 0.60 g/L during the first 30 days of lactation [46]. Furthermore, the level of all sialylated HMOs was more than two times lower in the milk from Swedish mothers (mean 1.6 mmol/L) than Ghana mothers (mean 3.6 mmol/L) [48]. Moreover, the concentration of 6′-SL and DSLNT were more than 4 and 2.5 times higher in milk from Ghana mother than Swedish mothers, respectively [48].

## 2. Concentration and Distribution of SMOs in Other Animal Species

SMOs are found not only in human milk but also in numerous other animal species. Thus, SMOs are as a key component in mammal milk to provide benefits to early life of all mammals. The purpose of this overview is to discuss the importance of SMOs in nursing offspring of different animal species and to provide updated information about alternative sources of SMOs for infant formulas.

### 2.1. Bovine Milk

Bovine milk contains a lower level of oligosaccharides than human milk at concentrations of 1–2 g/L and 0.05 g/L in colostrum and mature milk, respectively [26,49]. Although SMOs comprise approximately 70% of the total oligosaccharide present in the bovine milk [50], the absolute amount of total oligosaccharides is much lower in bovine milk compared to human milk throughout lactation. Therefore, the concentration of bovine SMOs is significantly lower than human milk, particularly 3′SL and 6′SL (Table 1). In mature bovine milk, total oligosaccharide concentration is about 6–7 times lower than mature human milk [27]. In addition to concentration, the structure and diversity of SMOs in bovine milk are also significantly less than human milk [18,51]. In human milk, 6′SL is the predominant SMO; however, in bovine milk, the concentration of 3′SL is higher than that of 6′SL (Table 1). Currently, 35 structures of SMOs have been identified in bovine milk [43]. Sundekilde et al. [52] documented that oligosaccharide concentration in bovine milk varies significantly by breed. For instance, milk from a Jersey breed had higher relative amounts of both SMOs and neutral fucosylated oligosaccharides; on the other hand, milk from a Holstein-Friesian breed contained a higher abundance of smaller and simpler neutral oligosaccharides. In a recent study, 13 oligosaccharides were identified in bovine milk, which were significantly influenced by breed, where most of these structures were abundant in the milk of the Jersey cattle compared to the Holstein-Friesian cattle [53]. Moreover, when parity of the cow was considered, it was found that the concentration of SMOs 3′-SL and 6′-SL, as well as neutral milk oligosaccharides (NMOs) lacto-N-tetraose, was higher in second-parity cows [53].

### 2.2. Caprine Milk

Caprine milk offers yet another potential alternative to human milk, due to the enhanced digestibility of goat milk compared to cow milk [67]. Goat milk is reported to be closer to human milk than bovine milk in regard to the protein, fatty acid, and casein content and structure of casein [68] and oligosaccharides [69]. The concentration of goat milk oligosaccharides is about 0.20–0.65 g/L in colostrum and 0.06–0.35 g/L in mature milk [69]. A total of 78 different structures of oligosaccharides have been characterized in caprine colostrum, with approximately 45% of these oligosaccharides being sialylated [70]. However, 64 oligosaccharides were identified in goat mature milk, of which 37 SMOs [71]. Same as in human milk, the predominant SMOs in goat milk is 3′SL and 6′SL [71]. While the overall milk oligosaccharide content of goats’ milk does not measure up to the quantities found in human milk, it is still noticeably higher than both bovine and ovine milk, with reports of quantities up to 5 and 10 times higher, respectively [60] (Table 1). However, cow’s milk-based infant formula is much more available and popular than goat milk-based infant formula. SMOs concentration in goat milk also varies according to the breed of goat. The concentration of total SMOs was a bit higher in Guanzhong than that in Saanen goat milk [71]. On the other hand, the abundance of 6′-SL was 3.3 times higher in the milk of Guanzhong goat than that in Saanen goat [71]. The Sia content of goat’s milk varies over the course of lactation, declining from approximately 0.47 g/L at 85 days of lactation to 0.12 g/L at 145 days of lactation [72].

### 2.3. Porcine Milk

One of the most translatable human infant models is the piglet, due to the prominent resemblance of their intestinal tract to humans [73]. Due to this resemblance, they are the preferred animal model for the human infant. Porcine milk contains relatively high milk oligosaccharides at concentration of 11.85–12.19 g/L and 6.82–6.98 g/L in colostrum and mature milk, respectively [74]. These levels are relatively lower than in human milk, but they are higher than in other domesticated dairy animals, e.g., bovine (0.03–0.06 g/L), goat (0.20–0.65 g/L), and sheep (0.02–0.04 g/L). A total of 94 milk oligosaccharide structures have been reported in porcine milk, of which 43 (46%) of them are sialylated [43]. Recently, we reported that porcine milk oligosaccharides (pMOs), including neutral-, sialyl-, and fucosyl-oligosaccharides in colostrum, were more abundant in the primiparous (gilt) than the multiparous female pigs (sow). SMOs account for approximately 58–78% of the characterized oligosaccharides in porcine milk, which is much higher than that of 10–30% found in human milk; however, the absolute concentration of pMOs is still much lower than human milk [43]. 3′SL is the dominant acidic oligosaccharide in porcine milk; similar to humans, the concentration of total SMOs, including 3′SL, decreases over the course of lactation [62,75].

Recently, we analyzed distribution and concentration of Sia from 8 gilts and 22 sows in colostrum, transition milk, and mature milk [76]. We found that gilt and sow milk contained significant levels of total Sia, with the highest concentration in colostrum (1.24 g/L), followed by transition milk (0.78 g/L) and mature milk (L 0.35 g/L). During lactation, the majority of Sia was conjugated to glycoprotein (41–46%), followed by SMOs (31–42%) and then gangliosides (12–28%). Neu5Ac was the major form of Sia (93–96%), followed by Neu5Gc (3–6%) and then KDN (1–2%), irrespective of milk fraction or stage of lactation. The concentration of Sia significantly declined during lactation [76]. We concluded that Sia glycans are important bioactive components that contribute to the optimization of neurodevelopment, immune function, and growth and development in piglets.

### 2.4. Elephantine Milk

Elephant milk contain significant amounts of oligosaccharides at level of 19–21 g/L in mature milk [63], which is significantly higher than that of any other species’ milk including human. Kunz et al.(1999) reported that total concentrations of oligosaccharides in elephant milk is three times greater than that of human transition milk, at concentration range from 10.60 g/L to 8.10 g/L at 45 to 234 days post-partum, respectively, determined by high-pH anion-exchange chromatography with pulsed amperometric detection and thin-layer cell (TLC) [63]. However, a total of 11 different structures of oligosaccharide were characterized, (with 4 of these being sialylated in the milk of 3 Asian elephants [63]. Furthermore, the proportion of SMOs is approximately 50% of the total oligosaccharides in elephant’s milk, which is higher than human milk [63]. The significant amount of SMOs in elephant milk may contribute to the higher complex memory and social behaviors of elephant than other non-primate species [77].

### 2.5. Equine Milk

The concentration of equine milk oligosaccharides (eMOs) ranges from 2.12–4.63 g/L, depending on the breed [64]. In total, 48 different structures of oligosaccharides have been characterized, of which 17% of them are similar to human milk [64,78]. A study utilizing thoroughbred mare milk during the first 7 days of lactation found that eMOs were predominantly neutral (58.3%), followed by Neu5Ac containing acidic oligosaccharides (33.3%) [78]. Neu5Gc were absent in equine mare milk [64], while Albrecht et al. [79] reported that Neu5Gc concentration was < 1% in mare milk, and 3′SL is predominant SMOs in mare milk [64]. It was also found that the structural profile of eMOs is most similar to porcine milk, followed by bovine, caprine, and then human milk (sharing 29, 28, 26, and 19 structures, respectively) [78]. Differences in presence and in abundance of specific eMOs are evident between breeds and within the breed [64].

### 2.6. Donkey Milk

There are few studies that exist on donkey milk oligosaccharides (dMOs); therefore, total concentration of dMOs has not been reported. Wang et al. (2019) reported that the concentration of SMOs in donkey milk is higher than that of neutral oligosaccharides [80]. The most abundant SMO and neutral oligosaccharide in donkey milk is 6’-SL and galactotriose, respectively [66,80]. The mean 6′-SL content in donkey milk is lower than human milk but higher than cow and mare milk (Table 1). So far, seven SMOs have been identified in donkey milk.

## 3. Sialylated Milk Oligosaccharides in Infant Formula Milk

World Health Organization (WHO) and United Nations International Children’s Emergency Fund (UNICEF) recommend that a newborn infant should be exclusively breast-fed, which means no other foods or liquids should be provided, including water, for the first 6 months of life. However, nearly 2 out of 3 infants are not exclusively breastfed for the recommended 6 months—a rate that has not improved in 2 decades [11,66,81]. Nevertheless, when breast-milk is not available, infant formulas offer a healthy alternative that attempts to mimic the nutritional composition of breast milk. Currently, the most common infant formula is based on bovine milk. In regards to fat, minerals, and protein, bovine milk contains higher concentrations compared to human milk [37,82]. However, the concentration and diversity of many other nutrients and biochemical components, such as SMOs, are significantly lower in bovine milk-based infant formulas [37,83]. This is because of the fact that cow’s milk contains significantly low diversity and concentration of SMOs (0.035–0.042 g/L) compared to human milk (2–3 g/L mature milk) [20]. Not much information are available on the accurate composition of SMOs in formula milk. Martin-Sosa et al. [83] reported that infant formula milks contain zero to negligible amount of SMOs. The concentration of Sia in mature human milk is 0.7 g/L, while bovine milk-based the formulas contain 0–0.2 g/L [37,84]. Therefore, formula-fed infants are estimated to receive only ~25% or less Sia compared to the exclusively breast-fed infants [37]. Furthermore, the vast majority of Sia in human milk are conjugated to MOs (73%), whereas, in cow’s milk-based formulas, it is mainly conjugated to glycoproteins (70%) [20,37]. Due to the diversity and concentration differences in SMOs between human and bovine milk, human milk exerts greater health benefits of protection against pathogens (Table 2) and improving neurodevelopment and cognition for infants compared to formula milk.

Moreover, the form of Sia is 100% Neu5Ac in human milk, which is about >25–80% higher than any commercially available infant formulas [37,81]. Human milk lacks Neu5Gc due to an inactivating mutation in the CMP-N-acetylneuraminic acid hydroxylase (CMAH) gene [90,91]. CMAH encodes for the rate-limiting enzyme in all cells responsible for converting Neu5Ac to Neu5Gc [92,93]. Milk Neu5Gc can be incorporated and metabolically into human tissues, since human biochemical metabolic pathways cannot discriminate Neu5Gc from Neu5Ac [94]. The Neu5Gc antigen known as xeno-autoantigen then leads to formation of anti-Neu5Gc antibodies, defined as xeno autoantibodies [95], which then can react with subsequent diet derived Neu5Gc [95]. The resulting antigen–antibody interaction is hypothesized to generate or promote chronic inflammation or “xenosialitis”, which is postulated to lead to greater risk of carcinomas in humans [94,96]. The concentration of Neu5Gc in cow’s milk-based infant formula is about 3–5% of the total Sia. Therefore, it is important to investigate the impact of formula milk-derived Neu5Gc on the health and wellbeing of newborns and infants, as well as on the long term health effect in adulthood [95,97].

## 4. Dose and Overall Functional Role of Sialylated Milk Oligosaccharides

Currently, the most commercially available forms of SMOs are 3′SL and 6′SL. In neonatal piglets, an ideal animal model for the human infant, 6′SL sodium salt supplementation to standard sow-milk replacer showed no adverse effects at the dose levels of 0.30 g/L, 0.60 g/L, and 1.20 g/L during 21 days study [98]. In neonatal rats, 3′SL sodium salt was tested for its safety at doses up to 5000 mg/kg body weight/day, and no toxicity or adverse effects was observed over the 90-day study [99]. While the concentrations of 3′SL and 6′SL are different, at current knowledge, they cover a range of biological functions in terms of pathogen resistance support [45], providing optimal conditions for the ‘beneficial’ intestinal bacteria [24], neural and cognitive development [12], intestinal maturation [100], and bone health [101]. Therefore, the inclusion of SMOs (isolated and purified from bovine and other animal species milk) in formula milk could close the gap between breastfeeding and formula feeding. Most recent studies of evaluating the functional role and underlying molecular mechanism of 3′SL and 6′SL in both animal and human health is summarized in Table 3.

## 5. Health Benefits of Sialylated Milk Oligosaccharides

### 5.1. Impact of SMOs on the Brain Development and Cognition

Brain developmental processes, ranging from neuroanatomy, neurochemistry, neurophysiology, and neuropsychology to long-lasting influences on cognitive events well into adulthood, heavily rely on nutrition during the first 2 years of life [116]. A large body of evidence shows breast-fed infants, particularly those born small or premature, grow up to have higher intelligence than children fed infant formula [117,118]. The question is: Why? The subject is controversial because it is difficult to disentangle genetic, environmental, and nutritional factors [118], and it is ethically unacceptable to conduct randomized controlled trials in human infants [119]. The question is one of profound clinical and public health importance; pre-term births, learning deficits, and behavioral abnormalities are increasingly common in children of both developed and underdeveloped countries [120]. Long chain polyunsaturated fatty acids (LCPUFAs), particularly docosahexaenoic acid (DHA), have been the focus of much research in this field [121]. However, there are other biochemical nutrients that have been shown to enhance neural development in animals [122]. We have been interested in HMOs, particularly SMOs and its key functional player Sia. Neural tissues utilize Sia as a key building block and human milk is one of nature’s richest sources; infant formulas contain little [123]. An exogenous source of Sia from SMOs may be critical under conditions of extremely rapid growth, e.g., the brain growth in the month after birth [124].

Currently, the detail metabolic fate of dietary SMOs or Sia in both animal and human body is not fully understood. Karim and Wang [30] speculated that high activity of neuraminidase in the intestinal mucosa of rat is related to a high level of sialylated molecules in the milk. After intravenous administration of isotope labeled free Sia in three-day old piglets, 0.23% of that Sia was located in the brain within two hours [18,125]. Oral and intravenous administration of radioactively labeled forms of both Sia and SL were found to be well absorbed (~ 90%) within 4 h by 20-day-old rat pups, 30% being retained in the body, and 3–4% in the brain, after 6 h [126]. It is important to note that current knowledge about mechanism of Sia incorporation into a newborn’s circulation before entering the nervous system are based on animal studies, which may not reflect the actual pathways utilized by humans [19].

Sia, as a marker of SMOs, play a crucial role in both brain development and the everyday functioning of neural cell membranes and their membrane receptors [127,128]. Of all the cells in the mammalian body, the neuronal cell membranes contain the highest concentration of Sia [124]. As a key component of sialylated glycoconjugates, such as polySia and gangliosides, Sia is involved in numerous critical aspects of brain development and functioning, such as synapse transmission, learning, memory, and cognition [129,130]. Thus, it has been hypothesized that adequate Sia intake is crucial for brain growth and development of the infant. A study conducted by our research team demonstrated that feeding a protein conjugated Sia during early development enhanced learning and increased expression of 2 genes associated with learning in developing piglets [131]. It was found that increasing Sia content in the diet significantly improved the ability for the piglets to learn the visual cue in the difficult tasks in a 8 arm radial maze [131], when 54 piglets were separated across four treatment diets varying in their Sia content for 35 days intervention. Moreover, the concentration of protein-conjugated Sia in the frontal cortex increased at a dose-dependent level [131]. Likewise, supplementation with 2 g/L of bovine 3′SL and 6′SL in pre-term pigs upregulated the total Sia in the corpus callosum and ganglioside-bound Sia was upregulated in a dose-dependent manner with 3′SL [16]. Recently, we provided in vivo evidences that milk 3′SL, 6′SL and 6′-sialyllactosamine (6′SLN) can alter many important brain metabolites and neurotransmitters required for optimizing neurodevelopment in piglets using in vivo magnetic resonance spectroscopic (MRS) approaches [30]. 3′SL and 6′SL have been shown to reduce anxiety-like behavior and maintain the levels of immature dentate gyrus neurons, when mice were exposed to a social disruption stressor and then subjected to open field and light/dark preferences trials [106].

Furthermore, dietary supplement of bovine milk oligosaccharides (bMOs )enriched whey with SL improved spatial cognition in a spatial T Maze test of pre-term piglets [102]. Additionally, genes involved in Sia metabolism myelination and ganglioside biosynthesis were found to be increased in the hippocampus of the SL-supplemented pre-term pigs [102]. Dietary Neu5Ac supplementation either in the form of Neu5Ac or 6′SL in rat pups for the lactation period improved cognitive and behavior performance at one year post-weaning and had better long-term potentiation (LTP) measured in the hippocampus compared to the control group [132]. Interestingly, between 3′SL and 6′SL, those rats consuming 6′SL had some enhanced cognitive outcomes, as well as increased polysialic acids-neural cell adhesion (PolySia-NCAM) expression in the frontal cortex [132]. Furthermore, dietary supplementation with SL or galactosylated *N*-acetylneuraminic acid (GN) tended to improve swimming results in a T-maze apparatus filled with water and in a Morris swimming-maze in adult rats [133]. Moreover, SL and GN intervention significantly increased the brain ganglioside concentration, and a trend of increased serum Sia in the rats. The SL and GN supplementation significantly upregulated the brain gangliosides GM3, GD1a, and tended to increase the GD1b and GT3 concentration in the rat [133]. However, dietary supplementation of Neu5Ac tended to upregulate brain gangliosides GM3 only, and GD2 was significantly higher in the GN diet compared to the control, lactose, galactooligosaccharide, and Neu5Ac diets, except SL [133]. These results indicate that dietary supplementation of SMOs enhances brain development, cognitive abilities, and brain sialylated lipid composition in rat.

### 5.2. Impact of SMOs on the Gut Microbiota and Necrotising Enterocolitis (NEC)

The SMOs as prebiotics reach the colon and act as a selective fermentation surface for “good” microbiota, such as *Bifidobacteria* and *Bacteroides*, promote their establishment, and aid in microbiota homeostasis [12,14]. These bacteria are then able to provide significant health and developmental benefits to the host by participating in establishing the early immune system, the production of key acids for biological functions, regulating metabolic and physiological function, as well as the gut epithelium and gut-brain axis [134,135]. Previous reports also indicate that addition of a Sia in cecal microbiota culture of piglets resulted in significant changes in the microbial community, such as relative rise in *Prevotella* and *Lactobacillus* species and reduction in the genera *Escherichia*/*Shigella*, *Ruminococcus*, and *Eubacterium* [136]. The utilization of SMOs by probiotic bacteria was first demonstrated by Idota et al., (1994), who showed that growth of *Bifidobacterium breve*, *Bifidobacterium infantis*, and *Bifidobacterium bifidum* were increased in the presence of SL [137]. Furthermore, *Bacteroides thetaiotaomicron* ATCC 2914, *Bifidobacterium longum* JCM 7007, 7009, 7010, 7011, 1272, 11347, ATCC 15708, *Lactobacillus delbrueckii* ATCC 7830, and *Bacteroides vulgatus* ATCC 8482 have improved growth on 3′SL and 6′SL treatment in vitro anaerobic cultures [24]. Additionally, SMOs are shown to increase the adhesion of *Bifidobacterium longum infantis* to HT-29 cells, with a mixture of both 3′SL and 6′SL significantly increasing adhesion, as well as simply 6′SL alone. Interestingly, 3′SL alone did not demonstrate any adhesion effects [138].

Studies have demonstrated that SMOs can alter the overall microbiota composition, as well as specific species [12,106,139]. Differences were shown in the microbiome of 21-day old piglets between the control and 6′SL treatment, in which the species *Collinsella aerofaciens*, the genera *Faecalibacterium* and *Ruminococcus*, and the genus *Prevotella* were increased in 6′SL-treated piglets [16]. Furthermore, the *Lachnospiraceae*, *Lactobacillales*, *Enterococcaceae*, and *Enterobacteriaceae* were decreased in 6′SL diet compared to the control diet [16].

The microbiota not only provides a source of nutrition to the infant, but it is also responsible for functions, such as the production of short chain fatty acids (SCFA); these fatty acids are well documented in aiding with gut barrier function and leukocyte function, as well as being a key energy source for intestinal epithelial cells [140]. SCFAs, such as butyrate, can be extremely beneficial for the animal, playing a role in reducing inflammation, pathogen population control, microbiota modulation, and gut development [141]. Furthermore, bacterial metabolites, such as indole, assist in establishing the epithelial barrier and have a protective role against dextran sodium sulfate (DSS)-induced colitis [142]. In vitro batch fermentation studies of the metabolism of both 3′SL and 6′SL by *Bifidobacterium infantis (B*. *infantis*), *Bifidobacterium bifidum* (*B*. *bididum*), *Bacteroides fragilis* (*Ba*. *fragilis*), and *Bacteroides vulgatus (Ba. vulgatus*) resulted in the production of lactate and SCFA, with acetate being the most highly produced SCFA [108].

While SMOs can increase the presence of probiotic species in the gut, they also can decrease the abundance of potential pathogens. Rotavirus, one of the leading causes of gastrointestinal illnesses and/or diarrhea in infants, can be managed by SMO [17]. Both in vitro and in vivo, SMO can inhibit Sia-dependent rotavirus infections by either inhibiting the binding of the rotavirus to the cells of the host or by preventing viral replication or entry [17]. Moreover, *Enterotoxigenic Esherichia coli* (ETEC), another leading cause of infant diarrhea is can bind with MOs, facilitating its passage out of the body and preventing attachment [143]. Human SMOs could inhibit hemagglutination of ETEC strains, whereas bovine SMOs were less effective at inhibition [143]. Cholera toxin, another cause of diarrhea, was shown to be inhibited by SL in a rabbit intestinal loop model [144]. Moreover, fluid accumulation resulting from cholera toxins, were shown to be minimized by SL [144]. Administration with SL, either 3′SL or 6′SL is able to aid in bacterial clearance of *Pseudomonas aeruginosa K*, a bacteria responsible for causing a wide range of diseases, such as respiratory, gastrointestinal, skin, and soft tissue infections, and upregulated the phagocytosis of macrophages [145]. The role of SMOs in infant growth, development, and health is summarized in Table 3.

Necrotizing enterocolitis (NEC), a fatal intestinal disorders in pre-term infants, occurs 6–10 times more frequently in formula–fed infants compared to breast-fed infants [20]. In a preclinical study, it was found that HMOs, such as disialyllacto-N-tetraose (DSLNT) and 2′-fucosyllactose (2′FL), reduce pathology scores and improve survival in neonatal rat model; however, DSLNT is most effective in preventing NEC [146]. Furthermore, GOS cannot exert this beneficial effect due to the lack of Sia in their structure, and modification of GOS by enzymatic incorporation of Sia (Neu5Ac) enables them to significantly lower the pathology score of NEC in neonatal rat [146]. Moreover, a multicenter human clinical cohort study showed that breast milk with lower concentration of DSLNT is more susceptible to develop NEC in low birthweight infants compared with the control; therefore, DSLNT content in milk can be used as a potential non-invasive marker to identify infants at risk of NEC [147].

## 6. Mechanisms via which Sialylated Milk Oligosaccharides Exerts Health Benefits

SMOs have been indicated to participate in regulating the health of the neonate by promoting intestinal maturation, regulating the microbiota, and improving cognitive abilities. SMOs are believed to be able to provide these health benefits through several different pathways. The proposed pathways for the effects of SMOs are outlined in Figure 1. Mechanisms by which SMOs impact human health need further heavily investigation.

## 7. Conclusions

The differences in the health status of breast-fed and formula-fed infants are not something that can be ignored, as we progress towards a world in which infants consuming formula as their sole or partial source of nutrition is increasing. Formula-fed infants have disadvantage scores in cognitive and behavior tests and have a higher incidence and severity of gastrointestinal illnesses. Human milk contains a significant high concentration of diverse soluble SMOs compared to any commercially available animal milk or infant formulas. However, milk from some animal species, such as elephant milk or goat milk, contain considerably higher concentration of SMOs. Therefore, more studies are needed to discover the potential health benefits of alternatives for human milk, but not limited to bovine milk, to ensure bottle-fed infants can receive same amount of nutrients as breast-fed infants. However, the mother’s milk of each animal species may be the best source of nutrition for their own progeny. Moreover, neutral oligosaccharides, such as 2′fucosyllactose, 3′fucosyllactose, have been extensively studied for their role in infant nutrition, growth, and development in both pre-clinic and clinic studies, but the benefits of SMOs are a little understood due to the fact that food-grade SMOs have not been fully developed. SMOs are a core component of breast-milk and play an important role in the growth and development of the brain, gut, and bone, as well as modulate immunity of newborn animals. It was documented that the activity of key bifunctional enzyme, UDP-N-acetylglucosamine-2-epimerase/N-acetylmannosamine kinase (Gne) for regulating synthesis of Sia [118] is lower in rat pups and guinea pigs and reaches maximum activity about day 15 [119]. Therefore, newborn human infants may have a lower capacity to synthesize Sia [18], which is a requisite precursor of brain gangliosides and sialylated glycoproteins [149], as well as a building block of SMOs. Thus, SMOs have been considered as an essential nutrients to neonates. Further studies of the health benefits of different structure SMOs in human milk are urgently needed.

## Figures and Tables

**Figure 1 foods-10-00473-f001:**
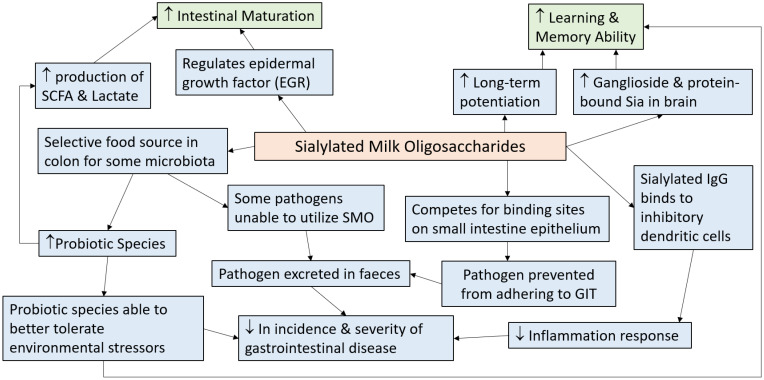
Summary of the pathways through which sialylated milk oligosaccharides (SMOs) exerts health benefits in humans and animals (adapted from References [16,35,100,103,106,108,109,114,148]). ↑ = increase, ↓ = decrease, GIT = gastrointestinal tract, IgG = immunoglobulin G.

**Table 1 foods-10-00473-t001:** Concentration of 3′sialylactose (3’-SL) and 6′sialylactose (6’-SL) in the colostrum and mature milk of different animal species.

	Colostrum (g/L)		Mature Milk (g/L)		References
	3′-SL	6′-SL	3′-SL	6′-SL	
Human Milk	0.09–0.35	0.25–1.30	0.17–0.50	0.17–0.50	[12,54,55,56]
Bovine Milk	0.09–1.25	0.03–0.24	0.04–0.12	0.01–0.09	[12,57,58,59]
Caprine Milk	0.18–0.25	0.10–0.13	0.03–0.09	0.05–0.07	[60,61]
Porcine Milk	0.09	N/A	0.01	N/A	[62]
Elephant Milk	1.89 (d45)	0.30 (d45)	0.86	0.34	[63]
Mare Milk	0.29–1.75	0.02–0.23	0.06–0.08	0.0004–0.0026	[64,65]
Donkey Milk	0.02 (d15)	0.02 (d15)	0.01–0.02	0.01–0.02	[65,66]

N/A = Data not available, d = days.

**Table 2 foods-10-00473-t002:** A summary of studies exploring the gut microbiota difference between breast-fed and formula-fed infants.

Study Type	Breast-Fed Infants	Formula-Feeding Infants	References
30 breast-fed vs. 60 formula-fed (30 for formula A and 30 for formula B) babies recruited before or right after birth and exclusive breast/formula feeding for more than 4 months	At 40 days old ↓ diversity ↑ *Bifidobacterium* ↑ *Bacteroides* ↓ *Lachnospiracea* ↓ *Streptococcus* ↓ *Enterococcus* ↓ *Veillonella* ↓ *Clostridioid* At 3 months old ↓ *Lachnospiracea* ↓ *Clostridioid*	At 40 days old ↑ diversity ↓ *Bifidobacterium* ↓ *Bacteroides* ↑ *Lachnospiracea* ↑ *Streptococcus* ↑ *Enterococcus* ↑ *Veillonella* ↑ *Clostridioid* At 3 months old ↑ *Lachnospiracea* ↑ *Clostridioid*	[85]
6 breast-fed (11–22 days old) vs. 6 formula-fed (14 to 36 days old)	↑ *Bifidobacterium*	↑ microbial diversity ↑ *Atopobium* ↓ *Bifidobacterium* ↑ *Bacteroides*	[86]
700 breast-fed infants vs. 232 formula-fed infants at 1 month of age	↑ *Bifidobacteria*	↑ *E.coli* ↑ *C.difficile* ↑ *Bacteroides* ↑ *Lactobacilli*	[87]
35 breast-fed vs. 35 formula-fed (28 to 46 days old)		↑ *Clostridium paraputrificum* ↑ *C.perfringens* ↑ *Bacillus subtilis* ↑ *C. clostridifforme* ↑ *Bacteroides vulgatus* ↑ *Veillonella parvula* ↑ *Lactobacillus acidophilus* ↑ *E. coli* ↑ *Streptococcus bovis* ↑ *S. faecalis* ↑ *S. faecium* ↑ *C. difficile* ↑ *C. tertium* ↑ *Pseudomonas aeruginosa*	[88]
16 breast-fed vs. 6 formula-fed infants at 3 months	↑ *Bacteroides* ↓ *Clostridium* XVIII ↓ *Lachnospiracea incerate sedis* ↓ *Enterococcus* ↓ *Veillonella*		[89]

↑ = increase, ↓ = decrease.

**Table 3 foods-10-00473-t003:** Summary of in vivo and vitro studies documenting the beneficial impacts of sialylated milk oligosaccharides (SMOs).

Species	Type of SMO	Effect	References
***In vivo studies***			
Piglet	3′SL and 6′SL	Enhanced T-maze performance Increased expression level of mRNA glial fibrillary acidic protein gene encoding Increased expression level of myelin basic protein Increased expression level myelin-associated glycoprotein	[102]
Piglet	3′SL and 6′SL	Upregulated Ki-67 expression in ileum crypts Increased width of ileum crypt Downregulated both the severity and incidence of diarrhea Upregulated mRNA expression of ST8Sia IV	[103]
Piglet	3′SL and 6′SL	Both SMO diets upregulated absolute myoinositol and glutamate + glutamine	[30]
Piglet	3′SL and 6′SL	Both 3′SL and 6′SL increased the ganglisoside bound Sia in the corpus callosum Ganglioside bound sialic acid in the cerebellum was increased in 3′SL group Altered the microbiome of the 6′SL group	[16]
Premature Ppiglets, newborn mice, and human intestinal explants with NEC	3′SL	Reduced apoptosis Reduced inflammation Reduced weight loss Reduced histological appearance of Neonatal Necrotizing Enterocolitis (NEC) in the intestine In human and mouse intestine, reduced toll-like receptor 4 signaling	[104]
Germ-Free mice colonized with microbiome of stunted 6-month- old	Bovine SMOs, mainly SL	Increased the volume and cortical thickness of the femoral trabecular bone Decreased osteoclasts	[101]
Mice	SL-deficient mother	Decreased microbial diversity	[105]
Mice	3′SL or 6′SL	Reduced anxiety-like behavior during stressor tests Prevented changes to the microbiotal diversity resulting from stress Maintained normal numbers of doublecortin (DCX) + immature neurons	[106]
In vivo mouse models of Lewis lung carcinoma, melanoma, and colon carcinoma cells	3′SL	Binds to vascular endothelial growth factor (VEGF) binding site to block downstream activation signal Inhibited angiogenesis on tumor tissues Diminished tube formation, migrations and actin filament arrangement in VEGF treated endothelial cells	[107]
***In vitro studies***			
In vitro microplate study	3′SL and 6′SL	Utilized by *B. infantis*, *B. bifidum*, and *Bacteroides vulgatus* Enhanced Bifidobacteria population in pH-controlled batch fermenter and lactate containing short chain fatty acid (SCFA) were produced	[108]
Human colon carcinoma Caco-2 cell lines	6′SL	Reduced the adhesion of *Eschericia coli*	[109]
In vitro and ex vivo experiments using mouse model	3′SL	Inhibited cartilage degradation Upregulated *CO12a1* production to promote cartilage regeneration Downregulated *Mmp3*, *Mmp13*, and *Cox2* expression	[110]
In vitro invasion assay using Adherent A549 cells	6′SL and 3′SL	6′SL reduced pneumocytes invasion of *Pseudomonas aeruginosa*	[111]
In vitro study using epithelial monolayers	3′SL	Reduced adhesion of Enteropathogenic *Escherichia coli* (EPEC), Enteroaggregative *Escherichia coli* (EAEC), *Shigella flexneri* (ATCC 12022), Diffuse Adhering *Escherichia coli* (DAEC), and *Salmonella typhimurium* (ATCC 14028) to intestinal epithelial Caco-2 monolayers	[112]
In vitro study using human chronic myeloid leukemia cell line K562	3′SL	Binds to SIGLEC-3 in human chronic myeloid leukemia K562 cells suppresses cytokine signaling 3 Induces megakaryocyte differentiation and apoptosis in chronic myeloid leukemia cells	[113]
Adult and infant human epithelial cell lines and fecal batch cultures	3′SL and 6′SL	Induced differentiation and epithelial wound repair Upregulated total SCFA production Increased abundance of *Bacteroides*, *Ruminococcs obeum*, *F. prausnitizii*	[114]
In vitro and in vivo mouse model of collagen-induced arthritis	3′SL	Reduced clinical scores and severity and incidence of arthritis Reduced paw swelling Reduced serum levels of inflammatory cytokines Reduced synovitis and pannus formation Suppressed cartilage destruction	[115]

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
