# Peer review of "Current Perspective of Sialylated Milk Oligosaccharides in Mammalian Milk: Implications for Brain and Gut Health of Newborns"

_foods, 2021, doi:10.3390/foods10020473_

Round 1

Reviewer 1 Report

The review and knowledge adressed in this manuscript is important for developers of infant formulas, and it has been interesting reading. The topic fits well with the focus of the journal and its Special Issue.

Still, I am left with some remaining questions about the current state of SMO in existing infant formulas. The manuscript could easily improve, adding more points for reflection, and identify the need for more research.

Detailed comments and suggestions for improvement are given in the attached review report.

----------

Main Impression

The review and knowledge adressed in this manuscript is important for developers of infant formulas. And the authors are stressing the urgent need to take this knowledge intoproduction. The article fits into the scope of the journals Special Issue, with its focus on nonbovinemilk and nutrition and health. It is an important contribution to the topic “Healthaspects of non-bovine milks (human and animal intervention studies to examine their impacton bone, mood, brain health, etc.)”

However, the review does little in collecting advices on how to increase SMO’s in the

formulas. Also, the reader is left with remaining questions about the current state of SMO in existing infant formulas.

Overall, the language in the manuscript suffers from too many statements, that leaves the reader with unanswered questions. The manuscript could easily improve, adding more points for reflection, and identify the need for more research.

Detailed comments and suggestions for improvement are given below.

ABSTRACT

L 27 Highlights the aim of this review to “critically discuss” the current state of knowledge and the biology of the role of SMO in human milk, infant formula milks and milk from several other species on gut and brain health. Please include more knowledge (if available) on infant formula milks, as that is the least covered in this manuscript. If this kind of knowledge is not available, it would be of interest to know why.

L 29 Please emphasis if the studied response is only discussed for human nutrition, or for the offspring of the “several other species”. Please make sure that this is clear all through the manuscript where discussed.

INTRODUCTION

The introduction is showing little critics against heavy statements regarding nutritional advantages of nursing. This is a topic that is heavily debated, and a review ought to cover both sides of the adressed topic. Are there any questions/doubts about the SMOs that needs more light in this respect?

L38 Please include what type of diabetes (I or II)

L38-40 Please divide sentence into two. Is there a link between malnourished children and their intelligence that should be explained here? Please include a reference

L42 Please remove dot before the references.

L42-44 Please rewrite the sentence in order to clarify if it is growth and development of the infant or its brain.

L44-46 Please rewrite to clarify what type of microbiota this sentence is referring to. Is it gut microbiota?

L66-68 WHY is SMO left out of infant formulaes? Please explain.

SIALIC ACID IS AN ESSENTIAL MOLECULE OF SIALYLATED MILK OLIGOSACCHARIDES

L71-83 As expressed here, there are many structurally different forms of Sia. But only 3 basic forms are described. What about the specific functions explained, are these related to Sia, regardless of its structural form?

L86-89 Identifies the different conjugations of Sia to other molecules, and hence explains why oligosaccharides is of most interest. This could preferably be adressed earlier in the manuscript.

SIALYLATED MILK OLIGOSACCHARIDES IN HUMAN BREAST MILK

L99-116. The change of concentration of different SMOs during lactations, would be easier to follow if expressed in a diagram. Please consider if that is possible. The diagram could also include the changes of Sia during lactation.

L102 There is a misspelling of the word “concentration”. Please correct.

L117 What is meant by “commercially available” forms of SMO? Please clearify.

L125-126 Inclusion of SMO in infant formulas are suggested, based on studies listed in the same paragraph. What type of milk should be the source of such SMO? Please elaborate.

CONCENTRATION AND DISTRIBUTION OF SMOs IN OTHER ANIMAL SPECIES

This chapter clearly shows the different contents of SMO in milk from several animal species.

What is the purpose of this overview? To show the importance of SMO in nursing, or to select sources for using SMO in production of infant formulaes from other sources that bovine milk?

Please include a sentence to explain the purpose of this overview.

L129-130 The sentence is inclomplete. Please complete.

L151-162 The paragraph indicates that goat milk is better for human infants than bovine milk, but does not mention any infant formula actually using goats milk. Could goat milk be an alternative rawmaterial for infant formulaes? Please elaborate

L177 What is pMO? Please explain abbreviations when introduced.

L210-211 The sentence is incomplete. Please complete.

L214-216 The sentence is too long. Please divide in two. Also there is a lack of comparison in the sentence. (higher concentrations…., than what?)

L218 Does this mean that infant formulas contain Neu5Gc conjugated SMOs? Please include relevant information

HEALTH BENEFITS OF SIALYLATED MILK OLIGOSACCHARIDES

L226-229 Please include references.

L234-236 Are there any data on how much SMO different infant formulas contain? Please add relevant information

L243-254 This paragraph contains 5 different references, and the work is referred to as “we” – it is a bit confusing. Are all these publications from the same group? Please clearify.

MECHANISMS VIA WHICH SIALYLATED MILK OLIGOSACCHARIDES EXERTS HEALTH BENEFITS

Why is this a separate chapter?

L321 Referring to Figure 1. Please include the references the figure is developed from.

CONCLUSION

The conclusion neatly sums up the importance of understanding how Sia provide health benefits for infants, and how crucial it is to perform further investigations. However, there is a lack of applicational recommendation, and suggestion for use. Please include.

L330 SMO’s are already identified in infant formulas, yet this statement claims that food grade SMOs are not fully developed. Please explain.

TABLES AND FIGURES

L688 Is there a letter missing in the word “though”? Please correct.

Figure 1 Parts of the text in the figure are outside their boxes. Please adjust the text, so it fits the figure.

Figure 1. What are “stressors”? Please explain

Table 1 is not referrend to in the text. Please include a reference to the Table where

appropriate.

Table 2 is referred to AFTER table 3 in the text. Please adjust the order of the tables to fit the order of appearance in the text.

L697 Make sure that the legend of Table 3 points out that health effects are identifed in the offspring of the animal from which the milk is collected. (no human nutrition)

Table 3 Many of the results presented do not distinguish between in vivo and in vitro studies.

Please update.

Table 3 I suspect parts of the table is missing in my version of the manuscript. Some words are inclomplete in column 3, and there are no references included. Please update.

Reviewer 2 Report

Overall

The objective of the review is to summarize the knowledge concerning concentration and diversity of sialylated milk oligosaccharides in human and different animal specifies and their impact on brain and gut health of newborns.

The review covers the aim of the journal and the subject investigated is of worldwide interest. However, it would benefit the reader if the authors will addressed some points and add additional information. There is a lack of detailed data concerning the impact of milk maturation and gestational age on the concentration of sialylated HMOs as well as concerning the interindividual variation of HMOs between mothers. Does the secretor/non-secretor status of mother influence sialylation?

The description of some original data is too lengthy and unnecessary. The authors should address some points and add additional information to the particular subchapters. Moreover, some conclusions are not based on scientific evidence and are speculations only (some issues need to be clarified or updated). On the other hand, there are no specific conclusions based on the data analyzed in relation to human milk and milk of different domestic animals. In addition, for some statements there are no relevant references or references provided are from 20 years ago. I believe the topic can be of interest to many, but the review must be written in concise and clear language.

Major points

Title

The title does not quite match the review content - please modified both: title and manuscript.

Abstract

 L 24

“Most infant formula absent HMOs.” The sentence in this section is not obvious. Please rephrase.

L27-27

This review aims to critically discuss the current state of knowledge of the biology and role of SMOs in human milk, infant formula milks and milk from several other species on gut and brain health.”

Unfortunately, the subchapter concerning sialylated oligosaccharides in milk formulas is missing.

In revised manuscript some important data are missing.

  • How does sialic acid enter into a newborn's circulation and then nervous system?
  • Why Authors did not take into account data concerning donkey and mare milk in the text of the manuscript (only in table). On the other hand, data concerning porcine milk are missing in the Table.
  • Why Authors selected only 4 relatively old studies for the Table 1. Moreover, Table 1 is not cited in the text of the manuscript.
  • What about the recent findings of Professor Bode concerning the impact of sialylated oligosaccharides on reduction of NEC – recent works are not included.
  • Autran CA, Schoterman MH, Jantscher-Krenn E, Kamerling JP, Bode L. Sialylated galacto-oligosaccharides and 2'-fucosyllactose reduce necrotising enterocolitis in neonatal rats. Br J Nutr. 2016 Jul;116(2):294-9. doi: 10.1017/S0007114516002038.
  • Autran CA, Kellman BP, Kim JH, Asztalos E, Blood AB, Spence ECH, Patel AL, Hou J, Lewis NE, Bode L. Humanmilk oligosaccharide composition predicts risk of necrotising enterocolitis in preterm infants. 2018 Jun;67(6):1064-1070. doi: 10.1136/gutjnl-2016-312819
  • Autran CA, Schoterman MH, Jantscher-Krenn E, Kamerling JP, Bode L. Sialylated galacto-oligosaccharides and 2'-fucosyllactose reduce necrotising enterocolitis in neonatal rats. Br J Nutr. 2016 Jul;116(2):294-9. doi: 10.1017/S0007114516002038
  • Ramani S, Stewart CJ, Laucirica DR, Ajami NJ, Robertson B, Autran CA, Shinge D, Rani S, Anandan S, Hu L, Ferreon JC, Kuruvilla KA, Petrosino JF, Venkataram Prasad BV, Bode L, Kang G, Estes MK. Human milk oligosaccharides, milk microbiome and infant gut microbiome modulate neonatal rotavirus infection. Nat Commun. 2018 Nov 27;9(1):5010. doi: 10.1038/s41467-018-07476-4.

L44-46

Many studies have indicated a difference in the gut health and microbiota between breast-fed and formula-fed babies and have credited this difference to differences in milk oligosaccharide concentrations [15-17].”

The statement is too general, please clarify.

L48-49

The complex free oligosaccharides are made up of a core molecule consisting of glucose (Glu), galactose (Gal), fucose 48 (Fuc), N-acetylglucosamine (GlcNAc), and sialic acid (Sia) residues [11,18].”

Not all HMOs are built from 5 monosaccharides, please clarify.

L51-52

“Over 200 different oligosaccharides have been characterized in human milk [19,20] varying from 3 to 22 simple sugars [21]”

According to  Urashima et al. [2018] 247 varieties of HMOs have been separated, of which 162 chemical structures have been characterized. I suggest that the following paper should be cited. Urashima et al.: Human milk oligosaccharides as essential tools for basic and application studies on galectins. Trends Glycosci. Glycotechnol. 30, SE51-SE65, 2018.

L64-66

“Although galacto-oligosaccharides (GOS) and fructo-oligosaccharides (FOS) are currently included in some infant milk products to mimic the biological effects of HMO based on their structural similarities to the core molecules of HMOs, both GOS and FOS are not present in human milk [28].”

It would benefit the reader to explain what GOS and FOS are structurally.

L67-68

“In particular, SMOs are left out of the infant formula [29].”

The reference [29] Wang et al. Concentration and distribution of sialic acid in human milk and infant formulas. 2001, is inadequate. There are more recent studies in the databases.

L70

Title of subchapter “ SIALIC ACID IS AN ESSENTIAL MOLECULE OF SIALYLATED MILK OLIGOSACCHARIDES” - it is evident by definition; the same for “… is a biomarker of SMOs” [L71]

L73-74

“These acidic monosaccharides are found attached to the cell surface as well as the surface of soluble proteins located at the end of sugar chains in higher animals and some microorganisms [32].”

This sentence must be rephrase. The acidic monosaccharides are not attached directly  to the cell surface.

L85-86

“Sia is found in variable concentrations in human breast milk with concentrations of approximately 5.76 and 5.04 mmol/L 85 in pre-term and full-term colostrum and 2.56 and 1.98 mmol/L in mature milk respectively [29].” [29] Wang, B, et al. Concentration and distribution of sialic acid in human milk and infant formulas. 2001.

The sentence and references should be updated.

L117-126

This paragraph should be modified and presented as separate subchapter.

L129-131

The references are missing.

L139-143

“In addition to concentration, the structure and diversity of SMOs in bovine milk are also significantly different from human milk [18]. In human milk, 6’SL is the predominant SMO, however, in bovine milk, the concentration of 3’SL is higher than that of 6’SL (Table 2). Currently 35 structures of SMOs have been identified in bovine milk [42]. Overall abundance and diversity of SMOs in bovine milk is lower than those of human milk [42,55] (Table 2).”

Please modified to avoid repetition.

L146-148

In a recent study 13 oligosaccharides was identified in bovine milk which were significantly influenced by breed, where most of these structures were abundant in the milk of Jersey cattle compared to Holstein-Friesian cattle [57]. Moreover when parity of the cow was considered, it was found that the concentration of numerous oligosaccharides was higher in second-parity cows. [57]”

Please specify which types of oligosaccharides.

L172 and L134

The values of oligosaccharides concentration provided for bovine milk are different - please clarify/verify. In addition, the concentration values should be given in uniform units.

L177-178

“Much like in humans, SL is the dominant acidic oligosaccharide and the total concentration of SMOs decreases over the course of lactation [66].”

Please clarify which form of SL.

L186-187

“We concluded that Sia glycans are important nutrients that contribute to the optimization of neurodevelopment, immune function, and growth and development in piglets.”

I have doubts about the use of the word "nutrients" in the context of sialic acid. Sialic acid or sialylated oligosaccharides are classify as bioactive molecule or bioactive components. In fact, they are not an energy source for piglets. This suggestion applies to the entire manuscript.   

L214-216; L219-220

“In regards to fat, minerals and protein, bovine milk contains higher concentrations, however many nutrients, such as SMOs concentration and diversity in infant formula are different from human milk [29].”[Wang et al., 2001]

Furthermore, the vast majority of Sia are conjugated to oligosaccharides, whereas in formulas it is mostly bound to glycoproteins [29].”[Wang et al., 2001]

Again, the sentences and references should be updated.

L217-218

“The different forms of conjugated Sia are also different in formula milk compared to human milk, e.g. human milk is lack of Neu5Gc conjugated SMOs.”

It would benefit the reader to explain the possible impact of Neu5Gc on the health and development of newborns and infants.

L223

The subchapter “Impact of sialylated milk oligosaccharides on the brain development and cognition” must be substantially improved to avoid repetition. Moreover, additional references must be added [L232-236].

Conclusion

There are no specific conclusions based on the data analyzed in relation to human milk and milk of different domestic animals.

Moreover, it may be appropriate to soften up the conclusion at the end of the paragraph and add a few words of explanation.

Additionally, [L332-337]

“Therefore, newborn human infants may have a lower capacity to synthesize Sia, which is a requisite precursor of brain gangliosides and sialylated glycoproteins [101] as well as a building block of SMOs.”

The references concerning “human infants may have a lower capacity to synthesize Sia” is needed.

Minor

The order of citation of the Tables in the text of the manuscript is incorrect. Moreover, Table 1 is not cited in the main text of the manuscript.

References

L164-165 [de Sousa et al., 2015] is not included in the reference list.

L185 [Marefa Jahan 2016] - please change to the appropriate item.

L499 [60. van Leeuwen et al.] is not complete

Round 2

Reviewer 2 Report

Comments to Authors

The manuscript has been significantly improved, however some points still need attention. Some suggestions were not taken into account and no adequate explanation was provided.

Additionally, the references should be verified and supplemented by relevant items.

Again

L67-68 “In particular, SMOs are left out of the infant formula [29].” The reference [29] Wang et al. Concentration and distribution of sialic acid in human milk and infant formulas. 2001, is inadequate. There are more recent studies in the databases.

Au: According to the reviewer’s suggestion more recent references has been added. Please see line 76.

R: Unfortunately, the relevant latest references are still missing. For example

  • Wylie AD, Zandberg WF. Quantitation of Sialic Acids in Infant Formulas by Liquid Chromatography-Mass Spectrometry: An Assessment of Different Protein Sources and Discovery of New Analogues. J Agric Food Chem. 2018 Aug 1;66(30):8114-8123. doi: 10.1021/acs.jafc.8b01042. Epub 2018 May 11.
  • Claumarchirant L, Sanchez-Siles LM, Matencio E, Alegría A, Lagarda MJ. Evaluation of Sialic Acid in Infant Feeding: Contents and Bioavailability. J Agric Food Chem. 2016 Nov 9;64(44):8333-8342. doi: 10.1021/acs.jafc.6b03273. Epub 2016 Oct 27.

Again

“Why Authors selected only 4 relatively old studies for the Table 1. Moreover, Table 1 is not cited in the text of the manuscript.

Au: Table 1 (which is now Table 2 in the revised manuscript) has now been cited in the text of the manuscript. Please see line 258

R: Table 2 [references 143-146] New references have not been added to the Table 2 and still need to be completed.

Again

L177-178 “Much like in humans, SL is the dominant acidic oligosaccharide and the total concentration of SMOs decreases over the course of lactation [66].”Please clarify which form of SL.

Au: The reference paper does not says about the form of sialyllactose (SL).

R: However, additional data are available from other studies, e.g.

[141] Mudd AT, Salcedo J, Alexander LS, Johnson SK, Getty CM, Chichlowski M, Berg BM, Barile D, Dilger RN. Porcine Milk Oligosaccharides and Sialic Acid Concentrations Vary Throughout Lactation. Front Nutr. 2016 Sep 8;3:39. doi: 10.3389/fnut.2016.00039. eCollection 2016.

Again

L214-216; L219-220 In regards to fat, minerals and protein, bovine milk contains higher concentrations, however many nutrients, such as SMOs concentration and diversity in infant formula are different from human milk [29].” [Wang et al., 2001]

“Furthermore, the vast majority of Sia are conjugated to oligosaccharides, whereas in formulas it is mostly bound to glycoproteins [29].” [Wang et al., 2001]

Au: The sentences and references have been updated. Please see line 245-259

R: The most recent appropriate references were not added, although the authors indicated that have been added.

Line 249-250

“This is because of the fact that cow’s milk as the main formula ingredient contains significantly low diversity and 249 concentration of SMOs (0.035-0.042 g/L) compared to human milk (2-3 g/L mature milk) [31] .”

R: The references provided by Authors [31 Wang et al., 2001] is incorrect. Please provide the relevant reference.

Line 276

"...the dose levels of 300g/L, 600g/L and 1200 g/L during 21 days study [53]."

In text added, the provided values are incorrect.

Line 280-282

Therefore, the inclusion of SMOs (isolated and purified from human, bovine and other animal species milk) in formula milk could close the gap between the two methods of infant feeding.”

R: The information provided requires clarification.

What is the point to isolate SMOs from human milk?

Isn't it better to provide whole milk as donor milk?

Do we have effective method to isolate significant amounts of SMOs?

Table 1

R: In addition to Bruggencate et al. [113], more references to human milk are required in Table 1.

References

Some references are duplicated:

[19] Wang, B.; Brand-Miller, J. The role and potential of sialic acid in human nutrition. European Journal of Clinical Nutrition 2003, 57, 1351-1369, doi:10.1038/sj.ejcn.1601704.

[45] Wang, B.; Brand-Miller, J. The role and potential of sialic acid in human nutrition. European journal of clinical nutrition 2003, 57, 1351.

[31] Wang, B.; Brand-Miller, J.; McVeagh, P.; Petocz, P. Concentration and distribution of sialic acid in human milk and infant formulas. The American journal of clinical nutrition 2001, 74, 510-530 515.

[81] Wang, B.; Brand-Miller, J.; McVeagh, P.; Petocz, P. Concentration and distribution of sialic acid in human milk and infant formulas. Am J Clin Nutr 2001, 74, 510-515.

[41] Wang, B. Sialic acid is an essential nutrient for brain development and cognition. Annu Rev Nutr 2009, 29, 177-222, doi:10.1146/annurev.nutr.28.061807.155515. 556

[42] Wang, B. Sialic acid is an essential nutrient for brain development and cognition. Annual review of nutrition 2009, 29, 177-222.

[43] Wang, B. Molecular mechanism underlying sialic acid as an essential nutrient for brain development and cognition. Adv Nutr 2012, 3, 465S-472S, doi:10.3945/an.112.001875 5603/3/465S

[104] Wang, B. Molecular mechanism underlying sialic acid as an essential nutrient for brain development and cognition. Advances in Nutrition 2012, 3, 465S-472S.